# The Exposome Approach in Allergies and Lung Diseases: Is It Time to Define a Preconception Exposome?

**DOI:** 10.3390/ijerph182312684

**Published:** 2021-12-01

**Authors:** Juan Pablo López-Cervantes, Marianne Lønnebotn, Nils Oskar Jogi, Lucia Calciano, Ingrid Nordeide Kuiper, Matthew G. Darby, Shyamali C. Dharmage, Francisco Gómez-Real, Barbara Hammer, Randi Jacobsen Bertelsen, Ane Johannessen, Anne Mette Lund Würtz, Toril Mørkve Knudsen, Jennifer Koplin, Kathrine Pape, Svein Magne Skulstad, Signe Timm, Gro Tjalvin, Susanne Krauss-Etschmann, Simone Accordini, Vivi Schlünssen, Jorunn Kirkeleit, Cecilie Svanes

**Affiliations:** 1Center for International Health, Department of Global Public Health and Primary Care, University of Bergen, 5020 Bergen, Norway; marianne.lonnebotn@uib.no (M.L.); ane.Johannessen@uib.no (A.J.); Gro.Tjalvin@uib.no (G.T.); Jorunn.Kirkeleit@uib.no (J.K.); cecilie.svanes@helse-bergen.no (C.S.); 2Department of Occupational Medicine, Haukeland University Hospital, 5021 Bergen, Norway; nils.jogi@uib.no (N.O.J.); torilmknudsen@gmail.com (T.M.K.); Svein.Skulstad@uib.no (S.M.S.); 3Department of Clinical Science, University of Bergen, 5021 Bergen, Norway; Francisco.Real@uib.no (F.G.-R.); randi.J.Bertelsen@uib.no (R.J.B.); 4Unit of Epidemiology and Medical Statistics, Department of Diagnostics and Public Health, University of Verona, 37134 Verona, Italy; lucia.calciano@univr.it (L.C.); simone.accordini@univr.it (S.A.); 5Department of Pediatrics, Haukeland University Hospital, 5021 Bergen, Norway; ingrid.nordeide.kuiper@helse-bergen.no; 6Institute of Infectious Disease and Molecular Medicine and Division of Immunology, University of Cape Town, Cape Town 7925, South Africa; matthew.darby@uct.ac.za; 7School of Population and Global Health, University of Melbourne, Melbourne, VIC 3010, Australia; s.dharmage@unimelb.edu.au (S.C.D.); Jennifer.koplin@mcri.edu.au (J.K.); 8Department of Obstetrics and Gynecology, Haukeland University Hospital, 5053 Bergen, Norway; 9Department of Pulmonology, Medical University of Vienna, 1090 Vienna, Austria; barbara.x.hammer@meduniwien.ac.at; 10Danish Ramazzini Centre, Department of Public Health—Work, Environment and Health, Aarhus University, 8000 Aarhus, Denmark; amlw@ph.au.dk (A.M.L.W.); kpape@live.dk (K.P.); vs@ph.au.dk (V.S.); 11Murdoch Children’s Research Institute, Melbourne, VIC 3052, Australia; 12Department of Regional Health Research, University of Southern Denmark, 5230 Odense, Denmark; signe.timm@rsyd.dk; 13Research Unit, Kolding Hospital, University Hospital of Southern Denmark, 6000 Kolding, Denmark; 14Research Center Borstel, Leibniz Lung Center, 23845 Borstel, Germany; skrauss-etschmann@fz-borstel.de; 15National Research Centre for the Working Environment, 2100 Copenhagen, Denmark

**Keywords:** exposome, preconception, intergenerational, transgenerational, asthma, allergies, lung function, lung health, epigenetics

## Abstract

Emerging research suggests environmental exposures before conception may adversely affect allergies and lung diseases in future generations. Most studies are limited as they have focused on single exposures, not considering that these diseases have a multifactorial origin in which environmental and lifestyle factors are likely to interact. Traditional exposure assessment methods fail to capture the interactions among environmental exposures and their impact on fundamental biological processes, as well as individual and temporal factors. A valid estimation of exposure preconception is difficult since the human reproductive cycle spans decades and the access to germ cells is limited. The exposome is defined as the cumulative measure of external exposures on an organism (external exposome), and the associated biological responses (endogenous exposome) throughout the lifespan, from conception and onwards. An exposome approach implies a targeted or agnostic analysis of the concurrent and temporal multiple exposures, and may, together with recent technological advances, improve the assessment of the environmental contributors to health and disease. This review describes the current knowledge on preconception environmental exposures as related to respiratory health outcomes in offspring. We discuss the usefulness and feasibility of using an exposome approach in this research, advocating for the preconception exposure window to become included in the exposome concept.

## 1. Introduction

Emerging research suggests that environmental exposures, long time before conception in parents and grandparents, may contribute to the development of allergies and lung diseases. This research is in its infancy. There is, on the other hand, a vast body of research on how environmental factors can more directly affect these outcomes in one generation and, in the last few decades, there has also been a major development in the scientific understanding of how exposures occurring during intrauterine life and early childhood may affect life-span respiratory health. This knowledge is, nonetheless, inadequate to fully understand the origins of asthma, allergies, and lung diseases. New research explores mechanisms by which exposure effects may be transferred across generations and act in concert with the genome through epigenetic modifications that may ultimately influence offspring health and disease [1,2,3,4]. Human studies suggest that the transfer of environmental exposure effects across generations may take place, with relevance for the development of respiratory health and disease (Table 1).

There is growing awareness that the traditional methods of exposure assessment fail to capture the full extent of environmental exposures and their impact on fundamental biological processes [5]. The human reproductive cycle spans decades, and germline precursor cells undergo important developmental stages from the early intrauterine period to maturation and reproductive capacity [6,7,8,9,10]. Hence, shortcomings related to assessing concurrent exposures and their temporal aspects might be even more challenging when addressing exposures occurring during the preconception period, as there is a long lag time between the exposures and the manifestation of health and disease in the offspring. Importantly, the progression from exposure to disease involves a continuum of external factors and measurable molecular and genetic events [11,12]. This continuum provides opportunities for the use of innovative technologies to comprehensively assess the impact of exposures on respiratory outcomes.

The exposome is defined as the cumulative measure of external exposures on an organism, and the associated biological responses, throughout the lifespan, from conception and onwards. This term includes exposures related to the environment, individual characteristics, such as prematurity and infections, lifestyle choices, behavior, and endogenous processes at work and in daily life [11,13,14]. Until recently, epidemiological studies using the exposome approach to investigate asthma and lung function have only considered exposures during the prenatal (intrauterine) and postnatal periods [15,16] or adulthood [17], but not during the time periods before conception. A need for a better understanding of the preconceptional origins of disease through the paternal exposome, in particular, has been pointed out by Soubry [18]. Importantly, Golding et al. (2019) have used an exposome approach to report the associations of increased fat mass in adolescence with both a history of the paternal grandmother smoking prenatally, and of the father starting to smoke regularly before puberty, demonstrating the utility of this method to assess the preconception stage in human cohorts [19]. Applying an exposome approach to disentangle the contributing roles of multiple exposures in the parental preconception period, as well as the temporal relationship between exposures and biological responses, is becoming increasingly critical in addressing the complexity of asthma, allergies, and lung health in offspring (Figure 1).

In this review, we describe the current knowledge on the association between environmental exposures during the preconception period and the future offspring’s respiratory health in humans, including asthma, allergies, and lung function (Table 1). Furthermore, we discuss the feasibility of using an exposome approach to characterize the complex environmental exposures in the preconception window. To be noted, literature on the time window shortly before conception, and on the optimizing conditions prior to conception and pregnancy, is beyond the aim of this review. Furthermore, human studies are the main focus of this review, since animal studies are rarely used for an exposome approach that aims to embrace a complexity of exposures. We identified an animal study that used an exposome approach to investigate lung health, but in a one generation setting [20]. Hence, with regard to preconception exposures, this review only includes a discussion of animal studies on preconception exposures in relation to next generation(s) lung health.

## 2. Preconception Exposures and Respiratory Health: Evidence from Multigeneration Studies on Humans

### 2.1. Smoking

Tobacco smoking in different exposure windows may cause malignant and non-malignant diseases of the respiratory system [21], and it is also generally accepted that intra-uterine exposure to maternal smoking plays a key role in the child’s subsequent health outcomes. However, there is an emerging understanding of the mechanisms by which cigarette smoking, also in the father, may have adverse effects on offspring and subsequent generations [22].

In 2005, Li et al. published an analysis of data from the Children’s Health Study in Southern California, showing that a grandmother’s smoking during the mother’s fetal period increased the asthma risk in her grandchildren, independent of whether the mother smoked herself or not [23]. The findings could suggest an effect of the grandmother’s smoking on the germline cells of her fetus—the cells that would eventually give rise to her grandchildren. This has later been confirmed in various cohorts, using different methodology, and different approaches for the definition of the grandmother’s smoking and the offspring’s asthma [24,25,26,27].

In addition to supporting the results on grandmother’s smoking, the role of the father’s smoking in early puberty (i.e., before age 15 years) on the respiratory health of future offspring has been demonstrated in concordant findings from three epidemiological studies: the European Community Respiratory Health Survey (ECRHS), Respiratory Health In Northern Europe (RHINE), and Respiratory Health In Northern Europe Spain and Australia (RHINESSA) cohorts [28,29,30]. In these international multigeneration cohorts, extensive offspring- and parent-reported information on respiratory diseases and preconception exposures in the previous generation (together with objective measurements of lung function) were collected in both the maternal and paternal lines. In a first analysis of the RHINE study [28], the odds of nonallergic early-onset asthma in offspring was found to be three times higher if fathers started smoking before they were 15 years old. Intriguingly, there was a significant effect modification by the paternal grandmother’s smoking—this association was only present if the paternal grandmother did not smoke. On the basis of the ECRHS database, the multigeneration effects of tobacco smoking on asthma phenotypes in offspring were further investigated [29], while using more advanced statistical methods [31,32]. Additionally, in this analysis, the fathers’ smoking in prepuberty was found to significantly increase the risk of nonallergic asthma in offspring, whereas the grandmothers’ smoking when the mother was in utero was associated with asthma with nasal allergies in grandchildren. The inclusion of one unmeasured confounder in the models, which could represent genetic confounding, had a limited impact on the estimated effect of grandmaternal smoking in pregnancy. Validation analyses from the same cohorts on information reported across two generations showed that recall bias is likely to have a limited impact on these results [33,34]. Three-generation causal associations of the smoking of grandmothers and grandfathers on the lung function of offspring within the paternal line were assessed in the ECRHS and RHINESSA cohorts [30], using a causal modeling approach [6,32,35]. The father’s smoking in prepuberty appeared to cause lower lung function in offspring, with a negative direct effect on the offspring’s forced expiratory volume in one second (FEV_1_) and forced vital capacity (FVC). The paternal grandmother’s smoking in pregnancy had a negative indirect effect (through unobserved biological mechanisms for which the father’s lung function in adulthood is an indicator) on the grandoffspring’s FEV_1_/FVC ratio, suggesting a second-generation increased risk of airflow obstruction. Probabilistic simulations showed that unmeasured confounding would only have a low impact on these estimates.

In summary, there are several publications using different analytical approaches and data sources, finding adverse effects on the respiratory health of offspring secondary to the grandmother’s smoking (most studies on maternal grandmothers) and to the father’s early-onset smoking. Asthma and asthma phenotypes are the most studied outcomes, and there is one study with lung function as the outcome. All the studies show the harmful effects of preconception smoking, with more asthma or lower lung function, both regarding the intrauterine and the male early-puberty exposure windows. The analyses show relatively strong estimates. For instance, the associations with the father’s early-onset smoking were comparable to, or even stronger than, the associations with maternal smoking in pregnancy. The adverse effects of the father’s smoking in prepuberty on the offspring’s asthma and lung function have alarming implications from a public health perspective, as smoking in boys aged 11–15 years has increased in Europe over recent decades [36].

With regard to the smoking effects across generations, there is substantial mechanistic literature to support the findings from epidemiological studies. We will not review this literature, only briefly mention that the adverse effects of the grandmother’s smoking in pregnancy, and the father’s smoking in prepuberty could be explained by epigenetic alterations (such as DNA methylation, histone modification, and small RNAs) in developing germ cells, as tobacco smoking may cause heritable modifications of the epigenome [37,38]. The heritable effects of the father’s smoking in prepuberty on the offspring’s respiratory health are biologically plausible [22]. This exposure window is a critical period for germ line development, which might imply higher susceptibility to the effects of tobacco smoking on the gametes’ epigenome [7,8,9,10].

Studies on animals have demonstrated that preconceptional exposure to paternal smoking regulates the spermatozoan microRNAs (miRNAs) and possibly affects the offspring’s body weight in early life [39]. In another murine model, the prenatal and postnatal exposure to nicotine of the parental generation and, therefore, the preconceptional nicotine exposure of the offspring, were correlated with lung function deficits in the parents and the offspring. DNA methylations and histone modifications in parental lungs and gonads were suggested to play a key role in mediating the observed effects. To explain the decrease in lung function of the offspring, the authors hypothesize a transmission via the germ cells, as parental mice showed altered DNA methylation in testes and ovaries, as well as increased H3 acetylation in testes [40]. Studies on epigenetic patterns in humans have identified an association of prenatal paternal smoking with a higher DNA methylation of the immune-regulating genes in the offspring’s cord blood, which correlated to the development of asthma in the child [41]. Furthermore, a link between prenatal maternal smoking, genetic variants, and DNA methylation, with airflow limitation and airway reactivity, has been described by Patil et al. [42]. Knudsen et al. found specific DNA methylation patterns in the whole blood of adult offspring related to the father’s smoking [38].

### 2.2. Occupational Exposures

It is well-known that the occupational exposure in adulthood to a multitude of allergens and irritants can give rise to asthma in the exposed worker [43,44]. A few studies have also suggested that maternal occupational exposures during pregnancy to low-molecular-weight agents (e.g., cleaning agents, and hair dressing and dentistry chemicals), can pose a risk for asthma in the offspring [45,46,47]. In a Swedish registry study, a number of parental occupations, including nursing, home help, and cleaning and cooking, were associated with a higher risk of hospitalization for childhood asthma among offspring [48]. To date, three human studies have investigated the impact of preconception parental occupational exposures on allergies and lung health in offspring.

In an analysis of the RHINE cohort, Svanes et al. found that the father’s self-reported welding before conception, over at least ten years, was associated with a doubled risk of nonallergic early-onset asthma in future offspring [28]. The findings are, to some extent, supported by a murine study, evidencing that the paternal intraperitoneal exposure to chromium (III), a component of welding fumes, induced lung cancer in female offspring [49]. Using an asthma-specific job exposure matrix (JEM) to define occupational exposures in the RHINE, ECRHS, and RHINESSA cohorts, Pape et al. investigated four groups of exposures (allergens, reactive chemicals, microorganisms, and pesticides) [50]. Maternal and paternal exposures to occupational agents limited to the preconception period were, in general, not associated with asthma in offspring. However, maternal exposure to allergens and reactive chemicals, both pre- and postconception, was associated with increased odds for early-onset asthma in offspring. Using the same JEM, Tjalvin et al. investigated the specific job exposure category, “Indoor cleaning” (cleaning products/detergents or low-/intermediate-level disinfectants), an exposure present in 21 different professions in the International Standard Classification of Occupations-1988 (ISCO-88), including cleaning, personal care work, nursing, and cooking. Maternal occupational exposure to indoor cleaning agents starting at preconception and continuing around conception and pregnancy, and/or after birth, was associated with offspring childhood-onset asthma and/or wheeze. Exposure starting around conception and pregnancy was also associated with increased childhood asthma and/or wheeze, while maternal occupational exposure starting after birth was not associated with offspring asthma outcomes [51].

In summary, there is some evidence that parental occupational exposures may influence the respiratory health in future offspring, and that the possibility of effects on future offspring needs to be considered in further research on the health effects of occupational exposures.

### 2.3. Environmental Exposures

Air pollution is among the main known environmental threats to respiratory health [52]. Exposure to air pollutants during childhood, adolescence, and adulthood is associated with an increased risk of asthma attacks, rhinitis, and low lung function [53], and limited recent research addresses the potential impact of such exposure on asthma and allergies in offspring.

A large study of kindergarten children from China assessed the associations between outdoor [particulate matter (PM_10_), sulphur dioxide (SO_2_) and nitrogen dioxide (NO_2_)] and indoor environmental factors (renovation and mold/dampness) during the preconception, prenatal, and postnatal periods, with allergies in the children. Several associations were found for prenatal/postnatal exposures. Regarding the preconception environment (one year before pregnancy), exposure to renovation was associated with rhinitis-like symptoms [54].

Kuiper et al. investigated the preconception exposure to air pollution in the childhoods and adolescences (0–18 years) of the parents in relation to asthma and rhinitis in their future offspring [55]. Childhood air pollution exposure was assigned to the parents from the Norwegian and Swedish study centers of the RHINESSA study, on the basis of residential address histories in the form of geocoordinates. Five air pollutants were investigated: NO_2_, PM_2.5_, PM_10_, black carbon (BC), and ozone (O_3_). Exposure to PM_2.5_ and PM_10_ in the mother’s childhood was associated with a higher risk of offspring asthma, and exposure to PM_10_ was associated with a higher risk of offspring hay fever. The father’s exposure to O_3_ was associated with more offspring hay fever; however, the father’s BC exposure was associated with less offspring asthma. The study suggests that parental air pollution exposures before the age of 18 years may increase the risk of asthma and allergies in children. Further studies are needed, and there is a need to disentangle the different air pollutants through improving the methodologies of multipollutant models.

A farm environment is considered a proxy of microbial diversity, and may also reflect lower air pollution, higher greenness, specific chemical exposures, etc. A range of studies find a lower risk for allergies in persons brought up on a farm, while Timm et al. investigated whether parental and grandparental farm upbringing was associated with asthma in offspring, in an analysis of the ECRHS and RHINESSA cohorts [56]. The reporting of a farm upbringing by family members was validated by Timm et al. [57]. No evidence of an association with the offspring’s asthma [hazard ratio (HR) 1.12, 95% CI 0.74 to 1.69] was found when comparing the offspring whose parents were not from a farm to those with both parents from a farm. Likewise, there was no association regarding grandparental farm upbringing and offspring asthma, neither in the maternal HR of 1.05 (95% CI, 0.67 to 1.65), nor in the paternal line HR of 1.02 (95% CI, 0.62 to 1.68). These null findings were consistent when stratified by the offspring’s own upbringing or by asthma phenotypes.

A murine study on water pollutants reported that the offspring of male mice exposed to the water pollutant, microcystin-leucine arginine (MC-LR), via drinking water before conception showed growth deficits and thickened alveolar walls, as well as collagen deposition in the lungs at postnatal day 180 [58]. Paternal sperm piwi-interacting RNAs (piRNAs) were mostly downregulated, and the predicted targets were involved in the regulation of the embryo implantation pathways. In addition, the analyses of 15 piRNA-related genes revealed that heatshock protein 90 α (Hsp90α) was downregulated in paternal testes. The lentiviral knockdown of Hsp90α in the testes of the fathers recapitulated the phenotype in F1, as it was observed after paternal exposure to MC-LR, suggesting a causal relationship. The authors propose that Wnt/b-catenin signaling is affected by dysregulated piRNAs and contributes to the abnormalities in the lungs of offspring.

Overall, an association of the parental exposure to air pollutants at age 0–18 years with offspring asthma was found in a solid study, in which exposure data were derived from the geocoding of residential addresses. One large study indicates a possible effect of indoor factors the year before conception, while a role for parental and grandparental farm upbringing seemed unlikely given the consistently negative findings.

### 2.4. Metabolic and Hormonal Exposures

Overweight and obesity are recognized as major risk factors for asthma in both childhood and adulthood [59,60,61,62,63], and numerous epidemiological studies have demonstrated the association of the mother’s excess weight just before and/or during pregnancy in relation to the offspring’s asthma [64,65]. Furthermore, excessive weight and obesity appear to be detrimental to lung function in both children and adults, regardless of asthma status [66]. Furthermore, emerging evidence suggests that there are mechanisms whereby the father’s metabolic environment before conception could also impact the health of his future children [4,22,67,68].

In the RHINESSA generation study, Johannessen et al. [69] investigated the effect of the parental overweight status on the offspring’s asthma, considering different susceptibility windows throughout the parents’ preconception lifespans. The potential mediating role of the offspring’s own overweight status was also evaluated. Overweight status, in both the parents and the offspring, was defined on the basis of a validated tool of nine sex-specific body silhouettes [70,71]. The authors found that the onset of the father being overweight by puberty (i.e., “voice-break”) was associated with an increased asthma risk in the future adult offspring. Similar findings were not identified for the father being overweight in the other investigated time windows, or for the mother being overweight in any time window. Mediation analysis showed that this effect was direct and not mediated through the offspring’s own overweight status. Using data from the Tasmanian Longitudinal Health Study (TAHS), Bowatte et al. [72] investigated the association between the body mass index (BMI) trajectories of parents in childhood and adolescence, and asthma in their offspring, as reported by the parents. The heights and weights of parents, at several time points, from ages 4–15 years, were available from school medical records, and four BMI trajectories were identified. The authors found that a high BMI trajectory in fathers in this age span was associated with a higher risk of asthma in their offspring. No such association was observed for the maternal BMI trajectories. Lønnebotn et al. [73] investigated the potential impact of parental overweight status, starting before or after puberty, on the adult offspring lung function (FEV_1_, FVC, and FEV_1_/FVC). The gender of the offspring was included as a potential moderator, and the adult height and being overweight in childhood were included as potential mediators in the pathway between the parents being overweight and the offspring’s lung function. Statistical models developed for causal inference, on the basis of observational data, were used for this analysis. The study showed that the father’s being overweight, starting before puberty, had a negative effect on the adult lung function, FEV_1_ and FVC, in their sons. The effects were partly mediated through the sons’ adult height, but not through the sons themselves being overweight. No causal associations were found between mothers being overweight and the offspring’s lung function.

Hormonal factors are closely linked with metabolic status. An analysis of the Tokyo-Children’s Health, Illness and Development Study (T-CHILD) found that the use of oral contraceptive pills (OCP) before pregnancy was associated with a higher risk of wheeze, asthma, and rhinitis in their children at 5 years of age. Importantly, a longer period of use of the OCP conferred a higher risk for wheeze and rhinitis in offspring [74]. An analysis from the Norwegian Mother and Child Cohort Study (MoBa) did not find associations between the maternal use of combined OCP (estrogen and progestin) in the year prior to pregnancy and asthma or wheezing in offspring before 3 years old, while the use of progestin-only OCP was weakly related to wheeze in offspring at the age of 6–8 months [75].

In summary, three human studies support the concept that the metabolic environment in male prepuberty might influence the respiratory health of offspring, and the role of the exogenous sex hormones before conception cannot be disregarded. Although extensive for some other outcomes, animal studies on the metabolic preconception environment, specifically related to the lung health of offspring are not available and would be useful to elucidating the specific mechanisms.

### 2.5. Infections and Immunity across Generations

It is well-established that maternal exposure to infections plays a role in determining the health of offspring and their risk of allergies. Offspring born to mothers with experimental influenza A virus infection had lower weights and length gains in the first weeks of life with reduced hematopoietic development and deviated pulmonary immune cell populations [76]. Straubinger et al. found that offspring from mothers who were pregnant during the T helper 2 (Th2) phase of a *Schistosoma mansoni* infection displayed increased ovalbumin-induced allergic airway inflammation compared to offspring from uninfected mothers, while offspring from mothers who were pregnant during the Th1 phase, or the regulatory T cell (Treg) phases of the infection were protected from such an allergy [77]. Potentially related to this is the demonstration that antihelminthic treatment during pregnancy may increase the likelihood of atopic disease in children [78]. Maternal helminth infection may lead to expanded regulatory T-cell populations in the offspring, which impairs their responsiveness to allergenic challenge [79], but maternal antihelminthic treatment leads to the loss of this regulatory environment. However, the transfer of regulatory immune components may not be the only driver of protection from allergies; for example, maternal interferon gamma (IFN-γ) (which is produced in response to numerous type 1 immunity-related infections) was demonstrated to protect against experimental allergies in offspring [80]. Furthermore, antigen transfer has been shown to occur from mother to child. The preconception intranasal exposure of wild-type (WT) dams to ovalbumin (OVA) led to a tolerance in offspring towards allergic airway inflammation [81,82], induced by maternal allergen transfer and the OVA uptake by fetal dendritic cells. Understanding the mechanisms for the impact of a mother’s infections in pregnancy on her offspring’s outcomes may be helpful when addressing the potential health effects in the offspring of parents with preconception infections—a field with very scarce human studies.

Jogi et al. suggest that the parental exposure to helminths might increase the odds of allergic manifestations in the offspring [83]. In this study of a Norwegian cohort, the timing of the exposure in relation to the offspring’s births could not be determined; however, the associations followed a pronounced gender-specific pattern suggestive of a preconception exposure effect. Stokholm et al. found that the use of antibiotics to treat maternal infections in the 80 weeks before and after the pregnancy (as well as during pregnancy) was associated with an increased risk of childhood asthma [84]. In this study, the preconception and intrauterine/postnatal effects could not be separated. Preconception infections altering the health of offspring is supported by a study from López-Cervantes et al. using Norwegian registry data that investigated the timing of parental tuberculosis infection in relation to the offspring’s birth year and the association with offspring asthma. The study revealed an increased risk of asthma in the offspring of parents who had a diagnosis of tuberculosis in childhood and later preconception, as compared to those whose parents had tuberculosis after they were born [85]. The authors theorize that tuberculosis-induced epigenetic reprograming might alter parental immunity, which could alter the offspring type 2 immunity characteristics.

These findings from human studies are supported by murine studies. With regard to helminth infections, there are studies suggesting that maternal helminth exposure prior to pregnancy may have enduring consequences on the health and immune responses of offspring, effects that may last well beyond childhood: one study found that preconception helminth infection may alter the offspring’s immunity and microbiome [86], and another study found that the maternal immune profile can be transferred via breastmilk to the offspring [87]. Breastfeeding has generally been shown to reduce the risk of allergies [88,89,90,91], including in the case of maternal hookworm infection, where breastfeeding was associated with a decreased incidence of eczema in the offspring [78]. This is, however, not always the case [92,93], leaving room for the possibility that breastfeeding may be both a protective and a risk factor under various circumstances. The novel perspective, supported by the study of Darby et al., is that the immunity transferred through breastmilk could be influenced by infections that occurred years before conception. The transfer of maternal commensal bacteria during birth through vaginal delivery has been shown to be important for the establishment of the offspring’s microbiome, which, in turn, may affect the offspring’s development of allergies and asthma [94,95]. Changes to the microbiome from preconception maternal infection, as demonstrated by Nyangahu et al. [86], could be critical to determining this microbiome, or could, by other pathways, influence the risk of allergies in offspring. Darby et al. demonstrated the maternal transfer of a robust type 2 immune bias, induced by a preconception maternal helminth infection, strongly imprinted on the offspring, and protective against similar helminth infection in the pups [87]. Supporting this concept, Ghosh et al. proved the transmission of CD8(+) T cells to pups via the foster nursing of previously immunized foster dams to *Mycobacterium tuberculosis* or *Candida albicans*. The T cells were identified as specific for antigens against these pathogens [96]. Several studies suggest that the transfer of noncoding RNA (ncRNAs) through breastmilk is plausible [97]. There are indications that such transfer, on the other hand, could predispose offspring to enhanced allergic inflammation under certain conditions [98,99]. Overall, preconception maternal immune exposures and infections might possibly alter the children’s risk to a range of diseases, including allergies, and the suggested mechanisms include the transfer to offspring of maternal immune cells [87,96], cytokines [100], altered microbiome [86], antigens [82], and genetic molecules [97]. Another plausible mechanism for the transfer of parental immunity to offspring is via the male line. Germline epigenetic changes in the preconception period have been illustrated by *Toxoplasma gondii* infection in male mice, in which the infection caused small RNA profile changes in their sperm and led to subsequent altered behavior in the offspring [101].

Given these results, we hypothesize that parental infections, even years before conception, might contribute to immunologic and epigenetic changes that could result in allergies in the next generation.

### 2.6. Miscellaneous Exposures

The genetic inheritance of asthma across generations is well-known; however, one may question whether disease activity may influence the germ cell environment or other factors (such as the microbiome) that, in turn, may affect the development of asthma and allergies in future offspring. Based on data from the ECRHS cohort, at different time points during the study participants’ reproductive age, Bertelsen et al. found that the parent-reported asthma and hay fever of offspring were more strongly associated with a parent’s preconception bronchial hyperresponsiveness and specific immunoglobulin E (Ig-E) levels than with the parental levels of these measures after the birth of the child [98]. In other words, parental disease activity, in terms of the immunological and clinical markers of disease, measured before conception, was more strongly related to the offspring’s allergic outcomes than parental disease activity measured after birth. The observed patterns are more likely explained by epigenetic inheritance than by a shared environment, which would have led to stronger associations with the postnatal disease activity of parents, or genetic inheritance alone, which would have led to equal associations with parental preconception and postnatal disease activity [102]. The authors questioned whether asthma treatment that reduces symptom intensity could also be beneficial with regard to the outcomes of offspring. In response to this question, a recent analysis by Banjara et al., using ECRHS data, investigated the association of the parental use of inhaled steroids during pre- and postconception periods, with parent-reported asthma in their offspring before the age of 10 years. A higher risk of asthma was found when the father (strongest associations) or the mother had used asthma medication before conception of the offspring [103]. While the use of asthma medication is highly correlated with the severity of the asthma, this study provided no indicatives that treatment might reduce the likelihood of asthma in offspring.

An analysis based on Swedish national registry data studied the association of maternal depression or anxiety at different periods (preconception, pregnancy, postnatal, or current) with asthma in her child. It evidenced that cumulative “exposure” to depression or anxiety was associated with offspring asthma, but the study was unable to identify a specific window when this association was stronger [104].

Altogether, these studies suggest that parental disease activity may need to be considered as potential risk factors for the respiratory outcomes of offspring, and that further research is needed.

## 3. Discussion

The reviewed literature strongly suggests that preconception exposures may be important for the development of asthma and lung health outcomes in future offspring. Associations with lung health outcomes of offspring are suggested for a range of single chemical exposures (smoking, occupation, and air pollution), hormonal metabolic exposures (being overweight and using oral contraceptives), and disease processes (infections, and asthmatic and allergic disease activity). In the context of preconception exposures, there is more evidence on the paternal line, and, in particular, for the childhood and early puberty exposure windows in males. The importance of these exposure windows may possibly be high; for instance, the associations between the father’s smoking starting before the age of 15 years and the offspring’s asthma risk appear to be at least as strong as for the associations with the mother’s smoking in pregnancy. Within the maternal line, associations with the asthma of a grandmother who smoked tobacco in pregnancy are reported in several studies. There is also some evidence of a maternal preconception exposure window in young adulthood.

The literature on the importance of preconception exposure for allergies and lung diseases in future offspring is supported by the evidence on preconception exposures for other health outcomes, such as obesity and growth [19,105]. Furthermore, the concept of preconception origins of health and disease is supported by mechanistic and experimental studies showing biological plausibility for the transfer of environmentally induced adverse impacts across generations. A role of the influence of germ cells, both for the female and male lines, is the most plausible pathway for the transfer of environmental effects across generations, and the mechanism most suitable for a discussion of the exposome effects, i.e., the effect of the totality of multiple concurrent or temporal exposures from all sources. One may speculate that the transfer of microbiome, or immune features influenced by the microbiome or infections, may become better documented and more relevant for future research on the preconception exposome.

### 3.1. The Exposome Concept

The exposome concept is divided into three different domains: the general external exposome, the specific external exposome, and the internal exposome [14,106,107]. The general external exposome includes the wider social, economic, and psychological influences (e.g., socioeconomic status, education, psychological and mental stress, urban and rural environments, etc.). This domain can partly be captured by questionnaires, registry data, and geocoding, while the use of personal apps and the global positioning system (GPS) to capture these domains is likely to increase in the years to come. The specific external exposome includes exposures such as chemical agents and environmental/occupational pollutants, infectious agents, radiation, noise, diet, lifestyle factors, and medication [107]. Traditionally, this type of information has been collected by questionnaires, expert-based exposure matrices, and through measurements of the agent in question in the ambient air, drinking water, or food. Loh and coworkers (2017) reviewed the possibilities and limitations in the use of wearable sensor technology and smart technologies as a means to measure the specific external exposure domain, including the use of the GPS for the measurement of personal location, which together with activity affect all three exposome domains [108]. Finally, the internal exposome includes chemicals or their metabolites in biological fluids and tissues, as well as nongenotoxic processes (metabolome, epigenome, transcriptome, proteome, adductome, and microbiome processes). In general, the high-resolution and high-throughput technologies needed for investigating the multiple omics above are already in place. A major challenge in the field of exposome research is that the biostatistical and bioinformatical methods needed for analyzing the vast number of combined omics and exposure data are lacking. This technology and knowledge are imperative in order to facilitate the translation and interpretation of the exposome to derive new biological knowledge that can be used in prevention and that has implications for human health. Even though the complexity will be even higher with data from two generations, methodological and technological advances should aspire to be able to also include the preconception window in the exposome concept.

### 3.2. What and When to Measure?

To have a beneficial or adverse effect on human health, the environmental exposure in question must interact with the biological system at some level, i.e., it must be bioavailable for the target cell at the right time and be able to interact with biomolecules. An optimal assessment of the exposome during the preconception period would require access to the germ cells of the parent and, perhaps, of previous generations, as well as (surrogate) target tissues in the next generation [6]. The human germ cells and the microenvironments of these in the ovaries/testes are unavailable for research in the prepuberty window, in both males and females. For the susceptibility window, from puberty to conception, it is possible to collect mature sperm and parts of its microenvironment from the paternal line. The sampling of egg cells (ova) from the maternal line would be excessively invasive to be feasible in large cohorts, and in relation to ovum retrieval, the microenvironment will most likely be affected by the treatment cycle prior to the procedure (e.g., hormonal and inflammatory effects).

The first obstacle for applying the exposome approach in the preconception period is the need to document the bioavailability and exposure intensities of the compounds of interest at the target cell. While the external exposome domain does not give us any information on the bioavailability of the compound, the endogenous exposome accounts for uptake from all sources (e.g., air, food, cosmetics, etc.) and routes (e.g., inhalation, oral, or dermal) of exposure. The use of the biological monitoring of exposure (chemical or its metabolites) in the assessment of health risk or prognosis requires that the biomarker is quantifiable, sufficiently correlated to the exposure of interest, and biologically relevant for the health effect or biological effect. Hence, although the internal concentrations of the environmental compounds of interest can be measured as proxies in accessible tissues (i.e., cord blood, blood, and urine) in future parents from birth and until conception, the true concentration available for the target cells or tissues (germ cells) during specific time points is not available.

For the same reasons described for bioavailability above, when judging whether it is biologically plausible that the human germ cells may be affected by preconception environmental exposures that also affect the immune and respiratory systems of future generations, we must rely on proxies of both exposures and effects, from existing and future human cohorts, combined with data from mechanistic and animal studies. The latter provides, for example, the possibility to gain detailed knowledge on how, and at what stages during testicular development and epididymal maturation, environmental information is transferred to the spermatozoa, and how the transfer of spermatozoan epigenetic information to the oocyte regulates transcriptional programs in the early embryo and the resulting later phenotypes. The experimental manipulation of epigenetic information, e.g., by injecting noncoding RNAs of interest into the early zygote, may potentially establish a cause–effect relationship.

### 3.3. Ethical, Legal, and Social Issues Related to the Use of Exposome Technologies

In parallel with the increased use of exposome technologies and the collection of high-resolution and high-dimensional data in health sciences, there is also a continuing need to address the ethical, legal, and social issues related to biobanking, archiving, and analysis of the large amount of biological material and data generated. These issues apply to all types of health research, but future offspring is a particularly vulnerable population. A main concern for discussion in large-scale epigenetic and genetic studies is the procedures for reporting pertinent and incidental findings back to the participants. This issue also applies to exposome research. At present, there seems to be a consensus that at least three requirements should be fulfilled prior to reporting research results: the results must be analytically valid, clinically significant, and actionable [109,110]. However, the criteria for what is valid, clinically significant, and actionable are not established for many of the exposures and outcomes that are being investigated nowadays, and must, hence, often be decided after a discretionary assessment.

For studies that investigate how the exposome at the preconception time window may affect future offspring, informed consent is provided by the parents or previous generations, but not by the subject at risk. This raises several ethical considerations for reporting back research results to the study participants and their offspring. How can we ensure that the offspring want to know whether he or she has an increased susceptibility to respiratory diseases in adult life, and how will this information affect future choices with respect to education and occupation, family life, or leisure time? Moreover, will his or her future employer or insurance company have the right to obtain this information before employment or before accepting him or her as a customer?

Finally, although there are analytical methods that allow examinations of thousands of chemicals in biological samples among the general population, the health risk for most of these chemicals cannot yet be determined. This hinders a liable risk communication back to the participants and, consequently, the biomonitoring of these chemicals is not performed. However, if we do not start collecting this type of biological material and information in ongoing and planned cohort studies for use in later agnostic and untargeted analysis, there will be important limitations in understanding exposure effects and interactions in the future.

### 3.4. Other Challenges

There are considerable challenges related to addressing the preconception environmental impact on the health of offspring in studies of humans, for whom the reproductive cycle spans over decades. Few cohorts have exposure data from parental prepubertal and pubertal years. Statistical methods for the optimal analyses of complex multigeneration data, with multiple mediators, confounders, exposures, and outcomes, in several generations, are becoming available, but there is a need for statistical expertise and methodology that also combine omics and exposure data representing the exposome across generations. Even with advanced methodology, the interpretation of negative results may pose a particular challenge, as they may reflect truly negative findings or a sum of limitations in complex data. Two partly negative papers have been published, showing an attention to publishing negative papers, but the publication bias of the statistically significant and stronger results is likely, as in other areas with less complexity.

## 4. Conclusions

Overall, the literature is sufficient to justify major research efforts on the preconception origins of asthma, allergies, and lung health. The research may have wide public health implications, indicating the potential benefit of interventions—or harm from negative exposures—for not only the exposed person, but also for future generations. Thus, it is urgent to bring this research forward. We argue that the benefits of an exposome approach in addressing complex, multiple, and concurrent lifestyles, behaviors, and exposures that interplay with each other may be equally important when considering the preconception environment as when addressing exposures during the life span. An exposome approach may be particularly useful to elucidating the germ cell environment at different developmental stages, from intrauterine life until the reproductive age of the parents. Exposure information from the preconception time window will, however, often be far more limited than for other life-stages, and the biomaterial from the preconception stages is generally not available, except for sperm, rarely available in humans and, if so, sampled after the offspring were born. This underscores the need for the close integration of epidemiological and mechanistic studies. Still, obtaining valid information about the preconception period should be kept in mind when planning questionnaires, the collection of biomaterials, the use of novel technologies, and data for validation analyses, in future inter- and transgenerational studies. We advocate for explicitly including the preconception period in the exposome concept in the quest to identify the environmental contributors to health and disease (Table 1).

## Figures and Tables

**Figure 1 ijerph-18-12684-f001:**
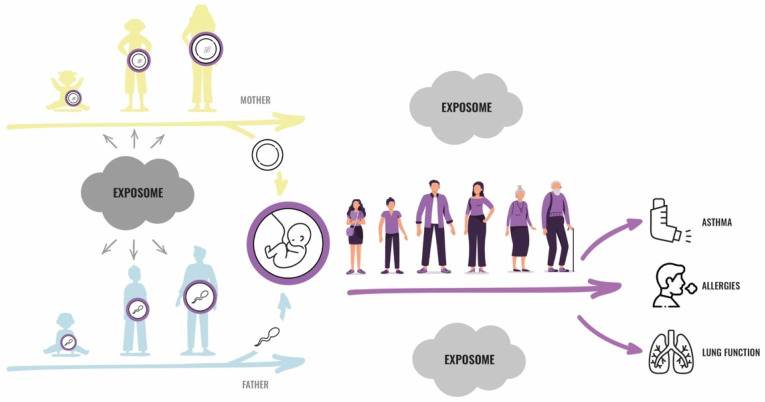
The preconception environment and the exposome approach. Maternal (yellow) and paternal (blue) preconception environments influencing germ cells (purple), and the possible impacts on offspring respiratory health, assessed through the exposome approach. (Nuria Báez Chocrón, Illus.).

**Table 1 ijerph-18-12684-t001:** Summary of multigeneration human studies associating preconception environmental exposures to asthma, allergies, and lung function.

Exposure	Outcome	Exposure Window	Main Findings	Study Cohorts ^1^	Reference
*Smoking*					
Smoking	Asthma	Grandmaternal pregnancy	Grandmother’s smoking during mother’s fetal period increased the risk of asthma in her grandchildren.	CHS	Li et al. Chest, 2005 [23]
Smoking	Asthma and wheezing	Grandmaternal pregnancy	Grandmother’s smoking during father’s fetal period increased the risk of asthma in the paternal daughter, in the absence of mother’s smoking during her daughter’s pregnancy.	ALSPAC	Miller et al. Chest, 2014 [27]
Smoking	Asthma	Grandmaternal pregnancy	Grandmother’s smoking during mother’s fetal period increased the risk of asthma in her grandchild, in the absence of the mother’s smoking during her offspring’s pregnancy.	MoBa	Magnus et al. Thorax, 2015 [26]
Smoking	Nonallergic early-onset asthma	Paternal prepuberty: paternal grandmother’s pregnancy	Father’s smoking in prepuberty increased the risk of asthma in his offspring, in the absence of grandmother’s smoking during the father’s fetal period.	RHINE	Svanes et al. Int J Epidemiol, 2017 [28]
Smoking	Allergic and nonallergic asthma	Paternal prepuberty; pregnancy	Father’s smoking in prepuberty increased the risk of nonallergic asthma in his offspring; grandmother’s smoking during mother’s fetal period increased the risk of allergic asthma in her grandchild.	ECRHS	Accordini et al. Int J Epidemiol, 2018 [29]
Smoking	Asthma	Grandmaternal pregnancy	Grandmother’s smoking during mother’s fetal period increased the risk of asthma in her grandchild, independent of the mother’s smoking during her offspring’s pregnancy.	NSC	Lodge et al. Clin Exp Allergy, 2018 [25]
Smoking	Persistent childhood asthma	Grandmaternal pregnancy	Grandmother’s smoking during pregnancy was related to an increased risk of early persistent childhood asthma in grandchildren. No risk for other asthma phenotypes was found.	Swedish national health registry-based cohort	Bråbäck et al. Pediatr Allergy Immunol, 2018 [24]
Smoking	Lung function	Paternal prepuberty; grandmaternal pregnancy	Father’s smoking in prepuberty reduced offspring’s FEV_1_ and FVC; grandmother’s smoking during father’s fetal period reduced the grandchild’s FEV_1_/FVC ratio.	Parents: ECRHS Offspring: RHINESSA	Accordini et al. Eur Respir J, 2021 [30]
*Occupational exposures*					
Welding	Nonallergic asthma	Paternal adolescence	Fathers’ preconception welding was associated with nonallergic asthma in offspring.	RHINE	Svanes et al. Int J Epidemiol, 2017 [28]
Allergens, reactive chemicals, microorganisms, and pesticides	Asthma	Before conception of child; pre- and postconception combined	Preconception maternal and paternal exposure to occupational agents was, in general, not associated with asthma in offspring. One exception was a higher risk of early-onset asthma if the mother had been occupationally exposed to allergens and/or reactive chemicals both before and after conception.	Parents: ECRHS Offspring: RHINESSA	Pape et al. Int Epidemiol, 2020 [50]
Cleaning products and disinfectants	Asthma and/or wheeze	Before conception of child; around conception and pregnancy	Mother’s exposure to indoor cleaning, starting before conception, was associated with offspring’s childhood allergic and nonallergic asthma, and/or wheeze.	Parents: RHINE Offspring: RHINESSA	Tjalvin et al. J Allergy Clin Immunol, 2021 [51]
*Environmental exposures*					
Outdoor pollutants and indoor new furniture/redecoration	Asthma and allergies	Before conception of the child	Preconception exposure to outdoor pollutants increased the risk of asthma and allergic rhinitis in childhood, while redecoration was associated with rhinitis-like symptoms.	CCHH	Deng et al. Chemosphere, 2016 [54]
Air pollution	Asthma and allergies	Parental childhood	Parental exposure to air pollution during childhood increased the risk of asthma and allergies in offspring.	RHINESSA	Kuiper et al. Int. J. Environ. Res. Public Health 2020 [55]
Farm exposure	Asthma	Parental childhood	Farm upbringing in previous generations was not associated with offspring asthma —either for parental or grandparental upbringing.	Parents: ECRHS/RHINE Offspring: RHINESSA	Timm et al. Int J Epidemiol, 2021 [56]
*Metabolic and hormonal exposures*					
Oral contraceptive pills	Childhood wheeze, asthma, and allergies	Before conception of child	Use of oral contraceptive pills increased the risk for wheeze, asthma, and rhinitis. Extended use of contraceptives increased risk for wheeze and rhinitis.	T-CHILD	Yamamoto-Hanada et al. Allergol Int, 2016 [74]
High BMI	Asthma	Parental childhood and adolescence	Father’s high BMI in childhood and adolescence associated with higher risk of asthma in offspring.	TAHS	Bowatte et al. J Allergy Clin Immunol, 2021 [72]
Overweight	Nonallergic asthma	Parental childhood, adolescence, and adulthood	Father’s onset of being overweight in puberty associated with offspring’s asthma without nasal allergies. The effect was independent of offspring’s overweight.	Parents: ECRHS Offspring: RHINESSA	Johannessen et al. J Allergy Clin Immunol, 2020 [69]
Overweight	Lung function	Paternal childhood/puberty	Father being overweight during childhood and/or puberty may cause lower lung function in offspring.	Parents: ECRHS Offspring: RHINESSA	Lønnebotn et al. Eur Respir J, 2021 [73]
*Infections and disease processes*					
Helminth infection	Allergies	Not known	*Toxocara spp* seropositivity in parents was associated with allergic outcomes in their offspring.	Parents: ECRHS Offspring: RHINESSA	Jogi et al. Clin Exp Allergy, 2018 [83]
Tuberculosis	Asthma	Parental childhood	Parental tuberculosis in childhood is associated to asthma in offspring.	Norwegian national health registries	López-Cervantes et al. Trop Med Int Health, 2021 [85]
*Miscellaneous exposures*					
Asthmatic and allergic disease activity (bronchial hyperresponsiveness and IgE levels)	Asthma and allergies	Before conception of child	Parental asthmatic and allergic disease activity measured before conception was associated to offspring asthma and hay fever.	ECRHS	Bertelsen et al. Clin Exp Allergy, 2017 [102]
Depression/anxiety	Asthma	Before conception of the child, pregnancy, postnatal and current	Cumulative exposure to maternal depression or anxiety was associated to asthma in offspring, but no specific period was found to be associated.	Swedish national health registries	Brew et al. Int J Epidemiol, 2018 [104]
Asthma medication	Asthma	Before conception of child	Parental use of asthma medication (inhaled steroids) before conception was associated with asthma in offspring.	ECRHS	Banjara et al. Trop Med Int Health, 2021 [103]

^1^ Study cohort: CHS: Children’s Health study in southern California; ALSPAC: Avon Longitudinal Study of Parents and Children; MoBa: Norwegian Mother and Child Cohort Study; RHINE: Respiratory Health In Northern Europe; ECRHS: European Community Respiratory Health Survey; NSC: Nationwide Swedish Cohort; RHINESSA: Respiratory Health In Northern Europe Spain and Australia; CCHH: China-Children-Homes-Health epidemiology study; T-CHILD: Tokyo-Children’s Health, Illness and Development Study; TAHS: Tasmanian Longitudinal Health Study.

## Data Availability

Not applicable.

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
