# Peer review of "The Exposome Approach in Allergies and Lung Diseases: Is It Time to Define a Preconception Exposome?"

_ijerph, 2021, doi:10.3390/ijerph182312684_

Round 1

Reviewer 1 Report

The theory of exposome is very interesting but to a large degree hypothetical.

It is well known that the exposition to toxic agents in men can be harmful for the spermatozoa. For example cigarette smoking is associated with higher level of DNA adducts in spermatozoa (but stronger correlation is with the male age!) as well as with higher DNA fragmentation index but this, in turn, impairs the fertility thanks to lowering the fertilization potential of spermatozoa, or may be associated with higher misscariage rate. It may be speculated that the DNA repairing system in the oocyte is not able to repair all the DNA damage of the spermatozoon but there is no direct evidence of any specific diseases in the offspring caused by this condition.

The DNA of the oocytes is resistant against the harmful effect of any agents from 16th week of pregnancy since from this time they are waiting in the prophase of the first meiotic division till the periconceptional period. The worse oocyte quality, as we know from the lessons of IVF, might be caused in this period by such internal factors as the obesity, polycystic ovarian disease or endometriosis influencing the hormonal function of the ovaries and/or milieu of the follicular fluid. So it is considering any external factors. Thus all these factors may impair the fertility of the women rather than cause any specific disease. It may be speculated that any traces of some toxic agents are persistent in female tissues (eg. heavy metals) and act on the embryo in the uterus after the conception. This, in turn, may lead to damage of the embryo, of course. Some theories such as that contraceptive use before conception may cause the lung disease in the offspring, may be socially harmful.

Finally, very often we cannot exclude the persistence of any harmuful conditions from preconceptional to postconceptional period (eg. smoking father still smokes in the presence of it’s pregnant wife).

So I propose the change of the title from „ The role of preconception environment in allergic and lung diseases: time to consider and exposome approach to characterizethe preconception environment” to  „ The hypothetical role of preconception environment in allergic and lung diseases: time to consider and exposome approach to characterize the preconception environment”

Author Response

Comment: The theory of exposome is very interesting but to a large degree hypothetical.

It is well known that the exposition to toxic agents in men can be harmful for the spermatozoa. For example cigarette smoking is associated with higher level of DNA adducts in spermatozoa (but stronger correlation is with the male age!) as well as with higher DNA fragmentation index but this, in turn, impairs the fertility thanks to lowering the fertilization potential of spermatozoa, or may be associated with higher misscariage rate. It may be speculated that the DNA repairing system in the oocyte is not able to repair all the DNA damage of the spermatozoon but there is no direct evidence of any specific diseases in the offspring caused by this condition.

The DNA of the oocytes is resistant against the harmful effect of any agents from 16th week of pregnancy since from this time they are waiting in the prophase of the first meiotic division till the periconceptional period. The worse oocyte quality, as we know from the lessons of IVF, might be caused in this period by such internal factors as the obesity, polycystic ovarian disease or endometriosis influencing the hormonal function of the ovaries and/or milieu of the follicular fluid. So it is considering any external factors. Thus all these factors may impair the fertility of the women rather than cause any specific disease. It may be speculated that any traces of some toxic agents are persistent in female tissues (eg. heavy metals) and act on the embryo in the uterus after the conception. This, in turn, may lead to damage of the embryo, of course. Some theories such as that contraceptive use before conception may cause the lung disease in the offspring, may be socially harmful.

Finally, very often we cannot exclude the persistence of any harmuful conditions from preconceptional to postconceptional period (eg. smoking father still smokes in the presence of it’s pregnant wife).

So I propose the change of the title from „ The role of preconception environment in allergic and lung diseases: time to consider and exposome approach to characterizethe preconception environment” to  „ The hypothetical role of preconception environment in allergic and lung diseases: time to consider and exposome approach to characterize the preconception environment”

Author response: Thank you for the useful mechanistic considerations. We agree that the understanding of the preconception environment as related to respiratory health is very limited, and we have modified the title as follows: The exposome approach in allergies and lung diseases: Is it time to define a preconception exposome?”. We believe this is carefully worded but still easy to read. If not acceptable, we are happy to use the exact wording indicated by the reviewer.

See attachment too.

Reviewer 2 Report

This paper is disappointing considering the skill and experience of the authors. This could be important but there is no survey of animal models which would seem to fit this project perfectly. If there are no animal models then this is an extremely weak area of research and if there are animal models why are they not discussed? This must be discussed for the paper to have any value. 

Author Response

Comment: This paper is disappointing considering the skill and experience of the authors. This could be important but there is no survey of animal models which would seem to fit this project perfectly. If there are no animal models then this is an extremely weak area of research and if there are animal models why are they not discussed? This must be discussed for the paper to have any value. 

Author response: In the revised version we have added relevant animal studies. An exposome approach with multiple exposures over long time is difficult in animal studies, but we have found one experimental exposome study in mice with lung health as outcome and have added this. Further, we have added animal studies investigating specific exposures in relation to lung health outcomes. The mechanistic literature about transfer of exposure effects across generations in general is, however, the subject of excellent reviews and beyond the scope of this review. An expert in animal studies, S Krauss-Etschmann, who could not find the time to contribute when the manuscript was first submitted, has now contributed importantly to adding animal literature to the revised version and is included as co-author. We have incorporated the revision of animal studies in the manuscript, which you can find as marked changes in the new version and below.

Further, human studies are the main focus of this review, since animal studies are rarely used for an exposome approach that aim to embrace a complexity of exposures. We identified an animal study that used an exposome approach to study lung health, but in a one generation setting [20]. With regard to preconception exposures, this review only includes a discussion of animal studies on preconception exposures in relation to next generation(s) lung health..” [Page 3, lines 109-114].

“In another murine model, prenatal and postnatal exposure to nicotine of the parental generation and therefore preconceptional nicotine exposure of the offspring were corre-lated with lung function deficits in parents and offspring. DNA methylations and histone modifications in parental lungs and gonads were suggested to play a key role in medi-ating the observed effects. To explain the lung function decrease in the offspring, the authors hypothesize a transmission via the germ cells as parental mice showed altered DNA methylation in testes and ovaries as well as increased H3 acetylation in testes [40].” [Page 5, lines 192-199].

“The findings are to some extent supported by a murine study, evidencing that paternal intraperitoneal exposure to chromium (III), a component of welding fumes, induced lung cancer in female offspring [49]” [Page 5, lines 218-220].

“A murine study on water pollutants reported that the offspring of male mice exposed to the water pollutant microcystin-leucine arginine (MC-LR) via drinking water before conception showed growth deficits and thickened alveolar walls and collagen deposition in lungs at postnatal day 180 [58]. Paternal sperm piwi-interacting RNAs (piRNAs) were mostly downregulated and predicted targets were involved in the regulation of the embryo implantation pathways. In addition, the analyses of 15 piRNA-related genes revealed that heatshock protein 90 α (Hsp90α) was downregulated in paternal testes. Lentiviral knockdown of Hsp90α in the testes of the fathers recapitulated the phenotype in F1 as it was observed after paternal exposure to MC-LR, suggesting a causal relationship. The authors propose that Wnt/b-catenin signaling is affected by dysregulated piRNAs and contributes to the abnormalities in offspring’s lungs.” [Page 6, lines 279-289].

“Although extensive for some other outcomes, animal studies on the metabolic precon-ception environment specifically related to offspring’s lung health are not available ,and will be useful to elucidate specific mechanisms.” [Page 7 and 8, lines 342-345].

“Offspring born to mothers with experimental influenza A virus infection had lower weight and length gain in the first weeks of life with reduced haematopoetic development and deviated pulmonary immune cell populations [76]” [Page 7, lines 348-351].

“The latter provides, for example, the possibility to gain detailed knowledge on how and at what stages during testicular development and epididymal maturation environmental information is transferred to spermatozoa, and how transfer of spermatozoal epigenetic information to the oocyte regulates transcriptional programs in the early embryo and the resulting later phenotypes. Experimental manipulation of epigenetic information, e.g., by injecting non-coding RNAs of interest into the early zygote may potentially establish cause-effect relationship.” [Page 11, lines 542-548].

“This underscores the need for close integration of epidemiological and mechanistic studies.” [Page 12, lines 610-611].

See attachment too.

Round 2

Reviewer 2 Report

this is a very good to great paper! thanks for adding the animal studies.